HFSA: hybrid feature selection approach to improve medical diagnostic system

Rabie Asmaa H. 1
http://orcid.org/0000-0002-5145-1907 Aldawsari Mohammed 2 mohammed.aldawsari@psau.edu.sa
Saleh Ahmed I. 1
Saraya M. S. 1
Rashad Metwally 2 3
1 Computer and Control Systems Engineering Department, Faculty of Engineering, Mansoura University , Mansoura , Egypt
2 Computer Engineering and Information Department, College of Engineering in Wadi Alddawasir, Prince Sattam Bin Abdulaziz University , Al Kharj , Saudi Arabia
3 Faculty of Computers and Artificial Intelligence, Benha University , Benha , Egypt
Alatas Bilal
Electronic publication date: 2025 May 6
Publication date: 2025
Volume: 11
Electronic Location ID: e2764
Received 2024 Dec 18; Accepted 2025 Feb 24
Copyright: © 2025 Rabie et al.
Copyright year: 2025
Copyright holder: Rabie et al.
License: This is an open access article distributed under the terms of the Creative Commons Attribution License, which permits unrestricted use, distribution, reproduction and adaptation in any medium and for any purpose provided that it is properly attributed. For attribution, the original author(s), title, publication source (PeerJ Computer Science) and either DOI or URL of the article must be cited.
License URL: https://creativecommons.org/licenses/by/4.0/

Keywords: Feature selection, Diagnosis, NB classifier, Optimization algorithm, Diseases, Artificial intelligence, Machine learning, Filter methods, Wrapper methods, Healthcare

Funding: Deanship of Scientific Research, Prince Sattam bin Abdulaziz University, Al-Kharj, Saudi Arabia PSAU/2024/R/1445 This work was supported by the Deanship of Scientific Research, Prince Sattam bin Abdulaziz University, Al-Kharj, Saudi Arabia via funding from Prince Sattam bin Abdulaziz University project number (PSAU/2024/R/1445). The funders had no role in study design, data collection and analysis, decision to publish, or preparation of the manuscript.

==============================
Thanks to the presence of artificial intelligence methods, the diagnosis of patients can be done quickly and accurately. This article introduces a new diagnostic system (DS) that includes three main layers called the rejection layer (RL), selection layer (SL), and diagnostic layer (DL) to accurately diagnose cases suffering from various diseases. In RL, outliers can be removed using the genetic algorithm (GA). At the same time, the best features can be selected by using a new feature selection method called the hybrid feature selection approach (HFSA) in SL. In the next step, the filtered data is passed to the naive Bayes (NB) classifier in DL to give accurate diagnoses. In this work, the main contribution is represented in introducing HFSA as a new selection approach that is composed of two main stages; fast stage (FS) and accurate stage (AS). In FS, chi-square, as a filtering methodology, is applied to quickly select the best features while Hybrid Optimization Algorithm (HOA), as a wrapper methodology, is applied in AS to accurately select features. It is concluded that HFSA is better than other selection methods based on experimental results because HFSA can enable three different classifiers called NB, K-nearest neighbors (KNN), and artificial neural network (ANN) to provide the maximum accuracy, precision, and recall values and the minimum error value. Additionally, experimental results proved that DS, including GA as an outlier rejection method, HFSA as feature selection, and NB as diagnostic mode, outperformed other diagnosis models.

Introduction

Nowadays, many dangerous and rapidly spreading diseases appeared that threaten the lives of people all over the world (Rabie, Saleh & Mansour, 2022; Rabie et al., 2022; Saleh & Rabie, 2023b; Sarker, 2021). Diagnosis of patients suffering from chronic diseases that are affected by newly emerging epidemics is very important to reduce the severity of vulnerability to the disease and the ability to follow up the case early and accurately (Saleh & Rabie, 2023a; Doi, 2007; Debal & Sitote, 2022). For this reason, it is important to find a modern diagnostic system to early and accurately diagnose patients to reduce the spread of diseases. This diagnostic system does not mean that the medical staff will be dispensed with, but it will be used to help them perform their duties, reduce their burdens, and also save waiting time for patients. Accordingly, determining a more accurate diagnostic system based on Artificial Intelligence (AI) methods is very necessary.

Many AI methods can be used as diagnosis methods in medical systems such as association rules, artificial neural networks, and Bayesian classifiers (Saleh & Rabie, 2023b). The applications of AI in medical systems, as provided in Fig. 1, include fore-casting the outcomes of diseases in the future based on historical data, preprocessing of missing data, analyzing treatment costs of resources, and disease diagnosis. Diagnosis several diseases take more attention from researchers to reduce the spread of diseases and also to reduce the burden on both medical staff and patients (Rabie, Saleh & Mansour, 2022; Rabie et al., 2022; Saleh & Rabie, 2023b). However, accurate diagnosis cannot be provided by using the current diagnostic models because of applying an unsuitable feature selection approach. Hence, it is a vital process to introduce an advanced selection method that can help the diagnostic system to give quick as well as more precise results.

Figure 1 Applying artificial intelligence in medical systems (Saleh & Rabie, 2023b).

This article summarizes the key contributions as follows: DS that includes three layers called (i) rejection layer (RL), (ii) selection layer (SL), and (iii) diagnostic layer (DL) will be introduced as a new system to diagnose cases accurately.

In RL, the genetic algorithm (GA) is used to correctly remove outlier data before training naive Bayes (NB) classifier.

In SL, hybrid feature selection approach (HFSA) is used as a new technique to accurately select the best features.

In fact, HFSA represents the main contribution in the diagnostic system (DS) in which it consists of two main stages called fast stage (FS) and accurate stage (AS).

Chi-square, as a filtering methodology, is implemented in FS to quickly select the best features.

Then, Hybrid Optimization Algorithm (HOA) that combines GA and Tiki-Taka Algorithm ( T2A), as a wrapper methodology, is applied in AS to accurately select the best features.

Based on the filtered data from both RL and SL, NB classifier is implemented in DL to give accurate diagnoses.

In fact, rejecting outliers by using GA and selecting the best features by using HFSA can enable the NB classifier to provide the maximum accuracy, precision, recall values, and minimum error value.

DS outperformed other diagnosis models.

The remainder of the article is organized as follows. The “Related Work” section reviews many previous works about feature selection techniques in medical diagnostic systems. Next, “The Proposed Diagnosis System” discusses the proposed diagnostic system. “Experimental Results and Discussion” describes the experimental results and discussion. Finally, “Conclusions and Future Works” includes the conclusions and future works.

Related work

Many previous works about feature selection techniques in medical diagnostic systems will be reviewed in this section. In Abdollahi & Nouri-Moghaddam (2022), a feature selection based genetic algorithm (FSGA) was provided as a new feature selection method to select the best set of features before training the ensemble classifier to accurately diagnose patients. Although FSGA can select accurate features, it needs a long execution time. Additionally, it was not validated on a dataset consisting of many various diseases. As mentioned in Rabie, Saleh & Mansour (2022), the Improved Binary Gray Wolf Optimization (IBGWO) model was provided as a new feature selection technique to select the optimal features before using the diagnostic method. Despite the advantages of IBGWO, its execution time is long. Additionally, it was only tested on the COVID-19 dataset but it was not validated on a dataset consisting of many various diseases.

As presented in Bashir et al. (2022), ensemble selection technique (EST) was introduced as a hybrid selection methodology applied to select the suitable features before using the diagnostic method. Although EST can select accurate features, it needs a long execution time. Additionally, it was not validated on a dataset consisting of many various diseases. In Rabie et al. (2022), hybrid selection methodology (HSM) was introduced as a hybrid selection method that was applied to select the suitable features before using the statistical NB method as a diagnostic method to detect cases suffering from COVID-19. Despite the advantages of HSM, it takes a long execution time. Additionally, it was only tested on the COVID-19 dataset but it was not validated on a dataset consisting of many various diseases.

In Ahmad & Polat (2023), the Jellyfish Optimization Algorithm (JOA) is a relatively new meta-heuristic algorithm inspired by the behavior of jellyfish in the ocean. Its application to feature selection has shown promising results, but it also comes with certain limitations. JOA is a simple algorithm with a few parameters to tune, making it easy to implement and computationally efficient. JOA can effectively identify the most relevant features, reducing dimensionality and improving model performance. On the other hand, JOA may converge prematurely to suboptimal solutions, especially for large-scale problems. The performance of JOA can be sensitive to the choice of parameters, such as the population size and control parameters.

As mentioned in Kadhim, Guzel & Mishra (2024), the Particle Swarm Optimization (PSO) algorithm has been applied as a wrapper feature selection method before applying the diagnostic model. PSO is relatively simple to implement and understand. It is robust to noise and can handle complex optimization landscapes. It can be easily adapted to different types of feature selection problems. On the other hand, PSO can be computationally expensive for large-scale feature selection problems. Incorporating complex constraints into PSO can be challenging, especially for feature selection problems with specific requirements.

According to Shokouhifar et al. (2024), a hybrid selection method that integrates Grey Wolf Optimization and Pearson Correlation Coefficient (GWO-PCC) has been used to select features before diagnosis patients. A common approach is to use GWO as a wrapper method to optimize the feature subset while using PCC as a filter method to pre-select a subset of potentially relevant features. This hybrid approach can leverage the strengths of both techniques, leading to improved feature selection performance. Although the high efficiency of GWO-PCC, it takes a large execution time.

Related to Jafar & Lee (2023), the Least Absolute Shrinkage and Selection Operator (LASSO) is a powerful technique for feature selection and regularization in diagnosis models. It works by adding a penalty term to the loss function, which encourages the model to shrink some coefficients to zero. This effectively removes irrelevant features from the model, leading to a simpler and more interpretable model. LASSO automatically identifies and eliminates irrelevant features, reducing model complexity and improving its generalization performance. The sparse nature of the LASSO model makes it easier to interpret and understand the impact of different features on the target class. Although these benefits, LASSO is sensitive to the scale of the features. It is important to standardize or normalize the features before applying LASSO to ensure that the penalty term has the same effect on all features. The performance of LASSO depends on the choice of the regularization parameter, which controls the strength of the penalty term. Finding the optimal value of this parameter often requires cross-validation, which can be computationally expensive.

Other many useful techniques have been used to improve medical systems. As highlighted in Sahu, Mohanty & Rout (2019), breast cancer is a leading cause of death among women, emphasizing the need for robust diagnostic tools. A hybrid feature selection method combined with principal component analysis (PCA) and artificial neural network (ANN) is introduced in Sahu, Mohanty & Rout (2019). While this hybrid approach shows promise for enhancing diagnostic accuracy, further research is needed to explore its limitations and optimize its performance on diverse datasets. In Sahu et al. (2022), a new hybrid feature selection algorithm that utilizes multiple filter-based rankers (MF) to efficiently reduce dimensionality and a hybrid political optimizer (PO) to effectively navigate the complex search space for optimal feature subsets was provided. By combining filter and optimization strategies, the method aims to balance computational efficiency with the ability to identify highly discriminative features, particularly for minority classes. It consumed a large time. As mentioned in Sahu & Dash (2024), hybrid information gain (IG) as filter method and Jaya Algorithm (JA) model as wrapper method for feature selection in microarray data was proposed to reduce the computational burden on the JA. IG-JA model achieved the absolute highest accuracy compared to all other approaches. The accuracy of IG-JA needs to be improved. According to Sahu & Dash (2023), the Grey Wolf Optimizer (GWO) with the JA as a hybrid wrapper selection method was proposed to mitigate GWO’s susceptibility to local optima stagnation. The added complexity of the hybrid approach (GWO-JA) compared to the basic GWO might increase the computational cost, despite potentially reducing the overall search time.

As highlighted in Sheikhpour et al. (2025b), a Semi-supervised Multi-label Feature (SMF) selection method was proposed to address leverage shared subspace learning to uncover underlying label correlations, effectively capturing the shared information across different labels. SMF has a high complexity. In Sheikhpour et al. (2025a), the Hypergraph Laplacian-based Semi-supervised Discriminant Analysis (HSDA) feature selection method was proposed, leveraging a trace ratio formulation. HSDA integrates hypergraph Laplacian to capture the geometrical structure and high-order relationships within both labeled and unlabeled data while maximizing class separability using labeled data. While HSDA effectively captures complex relationships and enhances feature selection through hypergraphs and sparse regularization, its computational complexity presents a potential disadvantage, especially for very large datasets. As presented in Saberi-Movahed et al. (2024), Non-negative Matrix Factorization (NMF) has become a prominent method in this domain, offering the advantage of producing non-negative components, which are often more interpretable in real-world applications like image processing and bioinformatics. While NMF excels in handling high-dimensional data and generating interpretable representations, it can be computationally expensive, particularly for very large datasets. The comparison between these recent models has been illustrated in Table 1.

Table 1 A comparison between the recent feature selection models.

Technique	Method type	Advantages	Disadvantages	
Feature Selection based Genetic Algorithm (FSGA) (Abdollahi & Nouri-Moghaddam, 2022)	Wrapper	It is based on the optimization algorithm. It is an accurate model compared to other models.	It takes a long execution time	
Improved Binary Gray Wolf Optimization (IBGWO) (Rabie, Saleh & Mansour, 2022)	Hybrid (Filter + Wrapper)	It is an accurate model compared to other models.	It was tested only on one disease (Covid-19). Slow	
Ensemble Selection Technique (EST) (Bashir et al., 2022)	Hybrid (Filter + Wrapper)	It is an accurate algorithm compared to other algorithms.	It was tested only on one disease (heart disease).	
Hybrid Selection Methodology (HSM) (Rabie et al., 2022)	Hybrid (Filter + Wrapper)	It is an accurate algorithm compared to other algorithms.	It was tested only on one disease (Covid-19). Slow	
Jellyfish Optimization Algorithm (JOA) (Ahmad & Polat, 2023)	Wrapper	It is easy to implement and computationally efficient.	It was tested only on one disease (heart attack). Its performance can be sensitive to the choice of parameters	
Particle Swarm Optimization (PSO) (Kadhim, Guzel & Mishra, 2024)	Wrapper	It is simple, robust, and easy in implementation.	It was tested only on one disease (heart attack). It is computationally expensive. Its complex constraints have challenges.	
Grey Wolf Optimization and Pearson Correlation Coefficient (GWO-PCC) (Shokouhifar et al., 2024)	Hybrid (Filter + Wrapper)	It has high efficiency.	It is slow	
Least Absolute Shrinkage and Selection Operator (LASSO) (Jafar & Lee, 2023)	Wrapper	It is easy to interpret and understand the impact of different features on the target class.	It is computationally expensive	
Principal Component Analysis (PCA) (Sahu, Mohanty & Rout, 2019)	Filter	It is fast and can enhance the diagnosis model.	It needs to be tested on different datasets.	
Multiple filter-based rankers (MF) and a hybrid Political Optimizer (PO) (Sahu et al., 2022)	Hybrid (Filter + Wrapper)	It can select the best features with high accuracy.	It takes a long execution time.	
Information Gain and Jaya Algorithm (IG-JA) (Sahu & Dash, 2024)	Hybrid (Filter + Wrapper)	It can reduce the computational of the JA.	This model needs to be improved by using more filter methods.	
Grey Wolf Optimizer with the Jaya Algorithm (GWO-JA) (Sahu & Dash, 2023)		It is accurate method.	It is slow.	
Semi-supervised Multi-label Feature (SMF) selection (Sheikhpour et al., 2025b)	Filter	It can effectively address label correlations and feature sparsity.	It has a high complexity.	
Hypergraph Laplacian-based Semi-supervised Discriminant Analysis (HSDA) method (Sheikhpour et al., 2025a)	Filter	It can provide accurate results according to pattern datasets.	It has a high complexity with large datasets.	
Non-negative Matrix Factorization (NMF) (Saberi-Movahed et al., 2024)	Filter	It accurately can handle high-dimensional data.	It is expensive.	

The proposed diagnosis system

Through this section, the diagnostic system (DS) is composed of three main layers; (i) RL, (ii) SL, and (iii) DL as shown in Fig. 2 will be discussed. The existence of outliers and irrelevant features will mislead the diagnostic method to give accurate results because of overfitting. Thus, removing both outliers and irrelevant features is important before learning the diagnostic method. To accurately remove outlier data, GA will be used in RL (Saleh & Rabie, 2023b). To correctly select the significant features, HFSA as a new method is used in SL. HFSA consists of two main stages FS and AS. Chi-square, as a filter methodology, is applied in FS to quickly select features while HOA, as a wrapper method, is applied in AS to accurately select features. Finally, the filtered data is passed to NB classifier in DL to give accurate diagnoses (Mahesh et al., 2022; Kumar et al., 2023; Umapathy et al., 2023; Ahsan, Luna & Siddique, 2022; Nasir et al., 2024). In this work, the main contribution represents HFSA as a new selection method. More details about RL, SL, and DL will be described in the next subsections.

Figure 2 The proposed diagnostic system (DS).

Rejection layer

Outlier rejection, also known as anomaly detection, is a data mining technique used to identify data points that deviate significantly from the norm (Rabie, Saleh & Mansour, 2022; Rabie et al., 2022; Saleh & Rabie, 2023b). In the context of medical diagnosis, these outliers can represent incorrectly recorded data due to equipment malfunction or human error, mistakes made during the input of patient information, or rare conditions or diseases that are not commonly observed. These outliers have a bad effect on the diagnostic model. Hence, outlier rejection is crucial for building accurate and reliable medical diagnosis models. Several methods are employed to detect outliers in medical data. These methods are categorized as statistical methods, clustering-based methods, and machine learning or optimization methods. while statistical and clustering-based methods are faster than optimization methods but optimization methods give more accurate results. In this work, GA is used as an optimization algorithm to give accurate results rather than its execution time because the outlier rejection process is performed only once and offline (Saleh & Rabie, 2023b).

When using GA for outlier rejection, the process typically involves the following steps. First, a population of candidate solutions is initialized. Each solution is a set of data points considered to be “normal” or representative of the inlier data. The remaining data points are implicitly treated as outliers. Next, an evaluation function (fitness function) assesses the quality of each solution. This function typically measures how well the selected data points fit a certain model or distribution, or how similar they are to each other, with better solutions receiving higher fitness scores. Then, selection operators choose solutions with higher fitness scores to become parents for the next generation, mimicking natural selection. Crossover operators combine parts of parent solutions to create new offspring solutions, exploring different combinations of data points. Mutation operators introduce small random changes to solutions, further diversifying the search. These steps of selection, crossover, and mutation are repeated for several generations, allowing the GA to evolve the population of solutions towards better representations of the inlier data. Finally, after a certain number of generations or when a satisfactory solution is reached, the algorithm outputs the best solution found, which represents the set of inlier data points. The data points not included in this final solution are then identified as outliers.

GA offers several advantages as an outlier rejection model, particularly when dealing with complex datasets where the nature of outliers might not be easily defined or where multiple types of outliers may exist. GAs can effectively search a high-dimensional feature space to identify optimal subsets of data points that represent the “normal” data distribution, implicitly flagging the remaining points as outliers. This flexibility is valuable compared to traditional statistical methods that often rely on assumptions about the data distribution, which may not hold in real-world scenarios. GA has been chosen for outlier detection because of its ability to adapt to complex data landscapes and potentially discover more nuanced outliers that other approaches might miss. While the computational complexity of GAs is a valid concern, especially with large datasets, it can be mitigated through various strategies. These include optimizing the GA parameters (population size, mutation rate, etc.), employing parallelization techniques to distribute the computational load, and incorporating domain-specific knowledge to constrain the search space. Furthermore, the offline nature of outlier rejection as a preprocessing step lessens the impact of GA’s execution time on real-time applications.

Selection layer

In this subsection, the proposed HFSA will be described in detail. In fact, medical diagnosis system is affected by irrelevant predictors (features) because these useless features mislead the model to provide accurate results and also cause over-fitting (Remeseiro & Bolon-Canedo, 2019; Ghaffar Nia, Kaplanoglu & Nasab, 2023). Hence, selecting the best predictors is a very important process before using the prediction model to give predication model the ability to provide correct predictions (Chae et al., 2024; Shivahare et al., 2024). Generally, there are two main categories of selection methods, namely; filter and wrapper (Rabie, Saleh & Mansour, 2022; Rabie et al., 2022; Saleh & Rabie, 2023b; Sarker, 2021). While filter type can speedily determine the most effective predictors, wrapper type can correctly determine the most effective predictors. Recently, wrapper methods based on bio-inspired algorithms have been used to correctly select the best predictors. During SL, HFSA is provided as a new method to select the best predictors that can correctly train the prediction model as illustrated in Fig. 3. Related to Fig. 3, HFSA includes two necessary stages; (i) FS and (ii) AS where FS is a filter stage that can quickly select a set of predictors and AS is a wrapper stage that can accurately select the best predictors. Consequently, all predictors are passed to FS to quickly select a set of predictors, and then the selected set of predictors from FS are passed to AS to accurately select the meaningful predictors.

Figure 3 The HFSA implementation steps.

According to Fig. 3, FS can quickly select a set of predictors based on the chi-square method (Saleh & Rabie, 2023a). After that, the selected predictors are passed to HOA as a new optimization algorithm in AS to accurately select the best predictors. In fact, HFSA is proposed to enhance the efficiency and accuracy of the diagnostic model. This HFSA employs a two-stage process. First, FS method utilizing chi-square analysis rapidly eliminates the least relevant features from the initial dataset. This pre-filtering step significantly reduces the dimensionality of the data before it is passed to the second stage (AS). The AS stage, which typically involves a computationally intensive optimization algorithm (HOA), benefits greatly from this reduction in feature space. By receiving a smaller, more promising subset of features from the FS stage, the AS process can more quickly and effectively identify the most significant features for accurate diagnosis, leading to improved model performance. Chi-square is chosen as the filtering methodology in the FS of the HFSA due to its computational efficiency and suitability for handling high-dimensional medical datasets. In such datasets, where the number of features can far exceed the number of samples, quick reduction of dimensionality is crucial. Chi-square excels at this by independently evaluating the relationship between each feature and the target variable (diagnosis). It assesses the statistical independence between them, effectively identifying features that are highly dependent on the diagnosis. This dependence suggests that the feature carries valuable information for distinguishing between different diagnostic outcomes.

Compared to other statistical methods, chi-square offers several advantages in this context. Methods like ANOVA, while powerful, assume the normality of data, an assumption often violated in medical datasets. Non-parametric alternatives like Kruskal-Wallis, while robust to non-normality, can be computationally more expensive than chi-square, especially with a large number of features. Furthermore, chi-square is specifically designed for categorical data, which is common in medical datasets (e.g., presence/absence of symptoms, genetic markers). While continuous features can be discretized, the method is inherently well-suited for many medical applications. Its computational simplicity allows for rapid filtering of features, making it ideal for the FS of the HFSA. While more sophisticated filtering methods exist, the goal of the FS stage is speed and sufficient reduction, not necessarily finding the absolute best subset. Chi-square offers a good balance between these competing priorities in the context of high-dimensional medical data.

The HOA in AS is indeed the basic method for accurate feature selection. However, its computational intensity makes it unsuitable for direct application to the original, high-dimensional dataset. Therefore, the chi-square-based filtering in the upper stage called FS serves as a crucial pre-processing step. By drastically reducing the number of features before they are passed to the HOA, the FS significantly curtails the HOA’s execution time. This allows the HOA to focus its computational resources on a smaller, more promising subset of features, ultimately leading to a more efficient and accurate feature selection process.

According to HOA in AS, it consists of two main algorithms, which are GA in a binary form (Saleh & Rabie, 2023b) and T2A in a binary form (Ab. Rashid, 2021). In a signal cycle of executing HOA, GA will be implemented on a set of predictors selected by FS and then new chromosomes in GA’s population will be passed as players in T2A’s population for implementation before starting to execute a new cycle. In other words, the output of GA passes into the input of T2A and then the output of T2A passes into the input of GA for the next cycle. To classify bio-inspired algorithms, there are two categories which are swarm intelligence and evolutionary (Davahli, Shamsi & Abaei, 2020). HOA includes the characteristics of both bio-inspired categories by using GA as an evolutionary algorithm and T2A as a swarm intelligence algorithm that mimics the football play behavior. GA can give the optimal global search that is related to its powerful in the exploration operation (Davahli, Shamsi & Abaei, 2020). GA can share the main information between chromosomes. At the same time, it can analyze many solutions. Also, it does not depend on mathematical derivation.

Although the advantages of GA, it cannot provide the best local search and it cannot give the optimal solution in the case if the probability of crossover and mutation are not suitable (Davahli, Shamsi & Abaei, 2020). Accordingly, GA can reach a local optimal solution but cannot reach a global one. To overcome the issues of GA and improve its performance, T2A is used because of its flexibility and scalability. Additionally, T2A that simulates the football play behavior can provide a high-speed convergence level (Ab. Rashid, 2021). Although the benefits of T2A, the performance of T2A has been affected by the change in the coefficient and also it relies on mathematical derivation. To overcome the issues of GA and T2A and utilize their benefits at the same time, HOA that includes both algorithms is provided to select the best predictors accurately.

The main reason for selecting GA and T2A is that GAs excel at exploring a wide range of potential feature subsets, effectively navigating complex search spaces to identify near-optimal solutions. Their strength lies in their ability to handle high dimensionality and non-linearity, crucial for medical datasets where feature interactions can be complex and the relationship between features and outcomes is often non-linear. However, GAs can be computationally expensive and may converge to local optima. T2A, inspired by the fluid passing style of football, offers a different approach. It emphasizes exploitation, efficiently refining promising solutions by iteratively improving upon them. T2A is generally faster than GAs and less prone to getting trapped in local optima due to its focused search strategy. The hybridization of these two algorithms capitalizes on their complementary strengths. The GA provides broad exploration, casting a wide net to identify potentially good feature subsets.

Then, the T2A algorithm takes these promising subsets and exploits them, fine-tuning the feature selection to achieve higher accuracy and efficiency. This combined approach addresses the limitations of each algorithm individually. The GA’s exploration helps T2A avoid premature convergence, while T2A’s exploitation accelerates the search process and improves the quality of the final feature subset. This synergistic combination is particularly beneficial for medical datasets, where identifying the most relevant features can significantly impact diagnostic accuracy and treatment effectiveness. The hybridization is needed because neither algorithm alone guarantees optimal performance. Specifically, these two methods were chosen because their exploration and exploitation strategies are highly complementary, offering a balanced approach that is well-suited for the challenges of feature selection in complex medical datasets.

There are many steps to execute HOA as illustrated in Fig. 3. Initially, it is proposed that the medical dataset consists of ‘u’ predictors. Based on this dataset, the chi-square method will be executed in FS to quickly select a set of predictors that consists of ‘v’ predictors, where v¡u. After that, the dataset that includes ‘v’ predictors will be transferred to HOA in AS to accurately select the best set of predictors that includes ‘m’ predictors, where m¡v. Hence, the dataset that includes ‘v’ predictors in FS will be passed to generate GA’s population to be executed. After that, a new population will be passed to T2A to be executed before starting the next cycle.

Initially, the execution of GA needs to create a population (PopGA) that includes ‘x’ chromosomes. The dimension of each chromosome is ‘v’ which represents the same number of predictors in the dataset (Rabie, Saleh & Mansour, 2022). Accordingly, the representation of each chromosome will be in a binary form including zero or one value. One value refers to the selected feature (predictor) but zero value refers to the unselected feature (predictor). Hence, a binary string representation (encoding) has been used where each bit corresponds to a feature in the medical dataset. A ‘1’ in the string indicates that the corresponding feature is selected, while a ‘0’ indicates that it is excluded. For example, if the dataset has 10 features, the string “1010011001” would represent a feature subset containing features 1, 3, 6, 7, and 10. This binary representation is simple, efficient, and commonly used in feature selection with GAs. After creating the GA’s initial population, NB’s accuracy will be calculated as an evaluation value to evaluate chromosomes using Eq. (1) (Rabie, Saleh & Mansour, 2022; Saleh & Rabie, 2023b).

(1) F(xi(t))=Accuracy_NB(xi(t))

where F(xi(t)) is a fitness value of ith chromosome at cycle t and Accuracy_NB(xi(t)) is an accuracy of NB related to ith chromosome at cycle t. In fact, this fitness function directly leverages the predictive power of a NB classifier. Each chromosome, representing a candidate feature subset, is evaluated by training NB model using only the features indicated by the ‘1’s in the chromosome’s binary string. The trained NB model’s performance, typically measured by its accuracy through cross-validation, directly forms the basis of the fitness score. Specifically, the fitness function can be defined as the NB model’s accuracy. This direct relationship between fitness and predictive performance ensures that the GA searches for feature subsets that maximize the NB classifier’s accuracy. In the next step, three operators called selection using the roulette wheel method, crossover using probability equals 0.9 and 1 bit method, and mutation using probability equals 0.01 and 1 flip bit will be used to produce new chromosomes in the population. In this study, two fit parents (chromosomes) are chosen at random using the roulette wheel approach, based on their probability of selection ((Psel) falling within the interval [0,1]; (Psel)∈[0,1]. Based on the probability of crossover (Pcross) pertaining to the interval [0,1]; (Pcross)∈[0,1], the crossover operator between the chosen parents is carried out using the one bit crossover method. Since crossover is a desired process, (Pcross) should be 0.9 or close to it. However, the mutation operator is carried out by flipping one bit according to the probability of mutation (Pmut) that belongs to the interval [0,1]; ((Pmut)∈[0,1]. Since mutation is actually a bad process, (Pmut) ought to be 0.01. Until the size of the new population matches the size of the original population, these three operators will be repeated.

These new chromosomes will be transferred to T2A as an input population (PopT) where each player includes one chromosome in binary form. After that, evaluating players will be performed by using the same fitness function of GA presented in Eq. (1). The best (key) player (q) will be assigned according to the evaluation values of players (Ab. Rashid, 2021). q will be modified in every cycle and also will be used as a global position to modify every player’s position through population. The ball’s position will be updated before the players’ position are modified using Eq. (2) (Ab. Rashid, 2021).

(2) pi(t+1)={r(pi(t)−pi+1(t))+pi(t),ifrandp>εpi(t)−(Ci+r)(pi(t)−pi+1(t)),ifrandp≤ε

where a new ball’s position is pi(t+1) in the next cycle t+1 while randp is a random value in (0,1). The basic idea of T2A depends on short passing which refers to moving the ball from the player to the nearby player. Losing the ball to the opponent may have occurred although the increase in the successful passing rate. For this reason, the probability of losing the ball is ε which has a percentage in [0.1–0.3]. In T2A, randp>ε represents the successful pass while randp≤ε represents the unsuccessful pass. C1 represents the ball’s reflection magnitude in the unsuccessful pass that takes values in [0.5,1.5]. (pi(t)−p(i+1)(t)) is the distance between ith position and ith+1 position of ball. In the case if pi(t) equals (pv(t)) that represents the last position of ball, p(i+1)(t) will be p1(t). After updating the ball’s position, every player’s position in PopT will be updated related to the ball’s position and the key player archive (q) using Eq. (3) (Ab. Rashid, 2021).

(3) pli(t+1)=pli(t)+r∗C2∗(pi(t)−pli(t))+r∗C3∗(q−pli(t))

where C2 is the coefficient that has value in [1,2.5] while C3 is the coefficient that has value in [0.5,1.5]. C2 and C3 are used to balance the player’s position (pl) between p and q. The representation of the new ball’s position and each player’s position includes continuous values. Hence, these continuous values should be converted to binary ones using a sigmoid function. Hence, the sigmoid function will be applied on the ball’s position; pi=(pi1,pi2,…..,piv) and every player’s position; pli=(pli1,pli2,…..,pliv) to convert them into binary form; pb_i=(pb1_i,pb2_i,…..,pbv_i) and plb_i=(plb1_i,plb2_i,…..,plvb_i) respectively using Eqs. (4) and (5) (Rabie, Saleh & Mansour, 2022; Rabie et al., 2022).

(4) pbiy(t+1)={1ifr(0,1)≥Sig(pbiy)0Else

(5) plbiy(t+1)={1ifr(0,1)≥Sig(plbiy)0Else

where the next cycle is t+1, the binary value of ith player for the next cycle at yth position is plby_i(t+1), and the ball’s binary value for the next cycle at yth position is pby_i(t+1). y represents a pointer to the current position (predictor); y=1,2,3,….,v. A random value that is belonging to [0,1] is r(0,1) while sigmoid functions for each player and the ball are Sig(plby_i) and Sig(pby_i) respectively which can be calculated by using Eqs. (6) and (7) (Rabie, Saleh & Mansour, 2022; Rabie et al., 2022).

(6) Sig(pbiy)=11+e−pbiy

(7) Sig(pbiy)=11+e−pbiy

where e indicates the base of the natural logarithm. Related to plby_i(t+1) as a new player’s position in PopT, new players in PopT are produced. Then, the new PopT is passed to PopGA as an input of GA. New chromosomes in PopGA will be evaluated using Eq. (1). These steps will be continued in the following cycles until the stopping criteria are satisfied. In the end, the optimal solution is represented in the chromosome that gives the maximum evaluation value where the best predictors are represented in all ones in this solution. Algorithm 1 presents the main steps of the HFSA algorithm.

Algorithm 1 Hybrid feature selection approach (HFSA) algorithm.

Input: IF: Input predictors in the medical dataset;   Train=(T,IF): Training dataset;   Test=(E,IF): Testing dataset;  u: Number of predictors in the medical dataset;	
  c: Number of chromosomes or players in population;	
  x: Number of chromosomes in GA’s population (Pop);	
   p1,p2,...,pl: Number of players in TA’s population (PopT);	
  Psel: Probability of selection;	
  Pcross: Probability of crossover;	
  Pmut: Probability of mutation	
Output: The best predictors in the fittest chromosome	
    Steps	
    Implement Fast Stage (FS)	
1: Select ‘v’ of predictors using the chi-square method.	
 Implement Accurate Stage (AS)	
2: Randomly create ‘c’ of chromosomes in GA’s population (PopG) in v-dimensions	
3: for each chromosome x∈PopG do	
4:  F(x)=Accuracy_NB(x)	
5: end for	
6: Select two chromosomes xi and xj based on Psel	
7: Apply crossover to xi and xj based on Pcross to produce new offsprings xi′ and xj′	
8: Apply mutation to xi′ and xj′ based on Pmut	
9: Enter xi′ and xj′ into new PopG as two new offsprings	
10: if no. of chromosomes in new PopG = pG then	
11:   Go to step 6	
12: else	
13:   Let PopT=newPopG; put players values in PopT equal new PopG	
14: end if	
15: Initialize the values of C1,C2, and C3	
16: for each player pl∈PopT do	
17:    F(pl)=Accuracy_NB(pl)	
18: end for	
19: Determine the key player (q) based on the maximum fitness value	
20: for each player pl∈PopT do	
21:    pl(t+1)=pl(t)+r∗C2∗(hl(t)−pl(t))+r∗C3∗(q−pl(t))	
22: end for	
23: for each position of the ball y∈V do	
24:    sig(py)=11+e−py	
25: end for	
26: for each player pl∈PopT do	
27:  for each position of the player y∈V do	
28:     sig(pl)=11+e−pl	
29:  end for	
30: end for	
31: for each position of the ball y∈V do	
32:    pb(t+1)={1ifr(0,1)≥sig(py)0else	
33: end for	
34: for each position of every player y∈V do	
35:    pl(t+1)={1ifr(0,1)≥sig(pl)0else	
36: end for	
37: Let PopG=PopT; put chromosomes values in PopG equal new PopT	
38: if the stopping criteria is not satisfied then	
39:   Go to step 3	
40: else	
41:   Return the set of predictors from the fittest chromosome that contains ‘m’ predictors; m≤v	
42: end if	
43: End=0	

Diagnostic layer

After implementing outlier rejection and feature selection processes, the medical diagnostic model uses NB classifier to analyze patient data and predict disease outcomes (Sarker, 2021). By removing outliers and irrelevant features, NB classifier significantly improves accuracy and reliability. Outliers, which are data points that deviate significantly from the norm, can distort the model’s learning process. Irrelevant features, on the other hand, can introduce noise and hinder the model’s ability to identify meaningful patterns. By eliminating these hindrances, NB classifiers can provide more precise and confident diagnoses, aiding healthcare professionals in making informed treatment decisions and ultimately improving patient outcomes.

Experimental results and discussion

During this section, the proposed DS which consists of three layers; RL, SL, and DL will be applied to accurately diagnose patients who suffer from different diseases. To implement DS, GA will be implemented at first to filter the dataset from outliers introduced in RL, then the proposed HFSA introduced as a new selection method in SL will be implemented to filter the dataset from irrelevant features, and finally, NB classifier as a diagnostic method in DL will be implemented to accurately diagnose patients. The proposed DS will be applied based on a dataset including several various diseases (Rabie & Saleh, 2024). For testing the performance of the used methods, confusion matrix metrics (accuracy, error, precision, and recall) will be used (Rabie, Saleh & Mansour, 2022; Rabie et al., 2022). Also, the false-positive and false-negative rates will be measured and visualized using the receiver operating characteristic and the area under the curve (ROC-AUC) (Saleh et al., 2025). Using K = 10, the most popular method of K-fold cross-validation will be applied, in which the dataset is split into k equal-sized folds. Consequently, the dataset has been split into one testing dataset and nine training datasets.

There will be two main scenarios used for the implementation. First, various current methods will be compared with the proposed feature selection approach (HFSA) using three different classifiers; NB, KNN, and ANN. Then, a comparison of the proposed DS with current diagnostic techniques will be performed in the second scenario. The values of used parameters are presented in Table 2.

Table 2 The values of the used parameters.

Technique	Parameter	Assigned value	
GA	Psel	Random value (0≤Psel≤1); psel=0.8	
	Pcross	Random value (0≤Pcross≤1); Pcross=0.9	
	Pmut	Random value (0≤Pmut≤1); pmut=0.01	
	r and rand	Random value belongs to (0<r,rand≤1)	
	Max_it	Maximum iteration number; Max_it = 200	
T2A	C1	The size of the ball’s reflected impact; C1=1.2	
	C2 and C3	The coefficients, C2=2.5 and C3=1	
	ε	The likelihood of losing the ball; ε=0.2	
	Max_it	Maximum iteration number; Max_it = 200	
KNN	K	The number of neighbors (1≤K≤15); K=7	
ANN	No. of hidden neurons	10	
	Training function	Levenberg-Marquardt backpropagation algorithm	
	No. of hidden layers	1	

Related to Table 2, the optimal values of the GA parameters, Psel, Pcross, and Pmut are 0.8, 0.9, and 0.01, respectively. Additionally, the best values of T2A parameters, C1,C2,C3, and ε are 1.2, 2.5, 1, and 0.2, respectively. Experimentally, the value of K is determined based on its value. Using a dataset of 500 examples, 400 of which are training data and 100 of which are testing data, KNN is implemented to determine the optimal value of K. Each value of K is used to calculate the KNN classifier’s accuracy and error. The ideal value of K is the one that helps KNN maximize accuracy while minimizing error. Throughout our study, K falls between 1 and 15; K ∈ [1,15]. Actually, K = 7 reduces the error rate, thus, it is the ideal value to employ in the next experiments. According to ANN, the training function is the Levenberg-Marquardt backpropagation algorithm. Additionally, the number of hidden layers includes ten neurons.

Dataset description

A medical dataset is a collection of historical data collected from patients with various illnesses (Rabie & Saleh, 2024). Based on 132 symptoms (features), 4,920 cases were identified in this dataset as belonging to 41 distinct illness categories. In other words, 41 distinct illness types can be identified using the 132 features in this dataset. Each characteristic in this dataset has a value of either “1” or “0,” which indicates whether or not the feature (symptom) has an impact on the illness. As a result, “1” denotes that the trait influences the disease, whereas “0” denotes that it has no effect. A total of 30% of the dataset’s cases are utilized for testing, while 70% of the cases are used for training. A training dataset of 3,444 cases and a testing dataset of 1,476 cases were therefore created from the 4,920 cases.

The complexity of DS

The DS’s complexity, or Big O, is primarily determined by two factors: the size of the training dataset (T) and the number of features (F). The size of the training dataset factor correlates with the training procedure’s O(T) complexity. The complexity of DS in respect to T and F can be calculated using Eq. (8) (Rabie & Saleh, 2024; Saleh et al., 2025).

(8) DS′scomplexity=O(T∗F)

Testing the proposed hybrid feature selection approach

The proposed HFSA will be evaluated to prove its efficiency against other feature selection methodologies after performing GA method as an outlier rejection method (Rabie, Saleh & Mansour, 2022). Then, three different classifiers, called NB, KNN, and ANN, will be used to validate HFSA compared to these previous methodologies (Rabie, Saleh & Mansour, 2022; Mahesh et al., 2022;). These selection techniques are FSGA (Abdollahi & Nouri-Moghaddam, 2022), IBGWO (Rabie, Saleh & Mansour, 2022), EST (Bashir et al., 2022), and HSM (Rabie et al., 2022). Figures 4–8 illustrate the evaluation of accuracy, error, precision, recall, and implementation time. In fact, HFSA outperforms other recent methods according to NB, KNN, and ANN.

Figure 4 Accuracy of selection techniques using (A) NB, (B) KNN, and (C) ANN.

Figure 5 Error of selection techniques using (A) NB, (B) KNN, and (C) ANN.

Figure 6 Precision of selection techniques using (a) NB, (b) KNN, and (c) ANN.

Figure 7 Recall of selection techniques using (a) NB, (b) KNN, and (c) ANN.

Figure 8 Implementation time of selection techniques using (A) NB, (B) KNN, and (C) ANN.

Figures 4–7 proved that the proposed HFSA is better than FSGA, IBGWO, EST, and HSM. HFSA introduces the best results at training patient number 3,444 according to all classifiers. Figures 4A–4C proved that HFSA is more accurate than other feature selection methods because it provides accuracy values for NB, KNN, and ANN that equal 89.91%, 85.85%, and 82.02% respectively. On the other hand, the error values of HFSA according to the same classifiers at the same order are 10.09%, and 17.98% as showed in Figs. 5A–5C. Hence, Fig. 4A showed that the accuracy of HFSA, FSGA, IBGWO, EST, and HSM according to NB are 89.91%, 62.21%, 73.92%, 75.48%, and 81.02%, respectively. Figure 4B showed that the accuracy of the same methods according to KNN are 85.85%, 61.01%, 71.82%, 73.40%, and 79.00%, respectively. According to ANN, Fig. 4C showed that the accuracy of the same methods is 85.00%, 61.00%, 71.02%, 72.42%, and 79.90%, respectively.

Figure 5A showed that the error of HFSA, FSGA, IBGWO, EST, and HSM according to NB is 10.09%, 37.79%, 26.08%, 24.52%, and 18.98%, respectively. Figure 5B showed that the error of the same methods according to KNN are 14.15%, 38.99%, 28.18%, 26.60%, and 21.00%, respectively. According to ANN, Fig. 5C showed that the accuracy of the same methods is 15.00%, 39.00%, 28.98%, 72.42%, and 20.10%, respectively.

Figures 6A–6C showed the precision of the proposed HFSA against other feature selection methods according to NB, KNN, and ANN. Figure 6A illustrated that the precision of HFSA, FSGA, IBGWO, EST, and HSM according to NB are 73.05%, 55.12%, 61.16%, 65.99%, and 71.05%, respectively. Figure 6B illustrated that the precision of the same techniques according to KNN are 72.00%, 54.01%, 59.18%, 64.00%, and 69.54%, respectively. Figure 6C illustrated that the precision of the same techniques according to ANN are 72.15%, 67.50%, 52.11%, 59.00%, and 64.02%, respectively.

Figures 7A–7C showed the recall of the proposed HFSA against other feature selection methods according to NB, KNN, and ANN. Figure 7A illustrated that the recall of HFSA, FSGA, IBGWO, EST, and HSM according to NB are 75.14%, 55.1%, 58.88%, 63.39%, and 69.47% respectively. The recall of the same methods in the same order according to KNN are 74.11%, 54.01%, 57.88%, 62.00%, and 68.40%, respectively as shown in Fig. 7B. Figure 7C showed that the recall of the same methods in the same order according to ANN are 73.00%, 54.55%, 57.01%, 62.55%, and 67.20%, respectively. According to these results, it is noted that HFSA represents the best feature selection method while FSGA represents the worst one according to NB, KNN, and ANN.

Experimental results demonstrate the superiority of the proposed HFSA, comprising Chi-square in FS followed by AS employing a hybrid algorithm (HOA) combining GA and T2A, over other feature selection methods, particularly in terms of diagnostic accuracy and precision. The HFSA consistently achieves higher performance metrics compared to methods like FSGA (which uses only GA for feature selection). This performance gap can be attributed to several factors. First, the initial filtering step in HFSA, by quickly eliminating irrelevant features, reduces the search space for the subsequent optimization stage, leading to more efficient exploration of the feature subset space. Second, the HOA within the AS stage likely provides a more robust search strategy compared to GA alone. GA excels at exploring the broader search space, while T2A helps refine the search around promising solutions, preventing premature convergence to local optima and improving the chances of finding the globally optimal feature subset. This combination of efficient pre-filtering and enhanced optimization contributes to the improved accuracy and precision of the HFSA.

Figures 8A–8C provided the implementation time of HFSA, FSGA, IBGWO, EST, and HSM based on NB, KNN, and ANN. In Fig. 8A, the implementation time of HFSA, FSGA, IBGWO, EST, and HSM, are 8.2, 8.68, 6.99, 5.89, and 4.8 s, respectively. In Fig. 8B, the implementation time of HFSA, FSGA, IBGWO, EST, and HSM, are 8.3, 8.90, 7.25, 5.91, and 4.99 s, respectively. In Fig. 8C, the implementation time of HFSA, FSGA, IBGWO, EST, and HSM, are 8.5, 8.91, 7.27, 5.98, and 5.25 s, respectively. Thus, the implementation time of HFSA is long but the implementation time of HSM is small according to NB, KNN, and ANN.

It is important to note that both outlier rejection and feature selection, although potentially complex and time-consuming preprocessing steps are typically performed offline before the diagnostic model is trained. This means their computational burden is not a factor during real-time diagnosis. Once the data is cleaned and the optimal feature subset is identified, the resulting model can be deployed for real-time use, providing rapid diagnoses based on the pre-processed, filtered data. Therefore, while the development time for these preprocessing steps is a consideration, their offline nature means their execution time does not directly impact the speed of real-time diagnosis.

Related to these calculations in Figs. 4–7, HFSA is the first best technique and HSM is the second best one as they can give the best results according to all classifiers. On the contrary, FSGA is the worst technique as it introduces the worst results. Additionally, the results depending on NB are better than the results of KNN and ANN. Consequently, DS based on GA as an outlier rejection method, HFSA as a feature selection method, and NB as a classifier method outperformed other systems that include GA as an outlier rejection method, different feature selection methods, and NB as a classifier. In the next subsection, the proposed DS depending on GA as an outlier rejection, HFSA as a feature selection, and NB classifier will be tested and compared to other systems.

Testing the proposed diagnostic system against other systems

The proposed DS that consists of GA, HFSA, and NB classifier will be evaluated to prove its efficiency against other diagnostic systems. In this work, the proposed DS will be tested and compared to Statistical Naıve Bayes (SNB) (Rabie et al., 2022), Hybrid Diagnosis Model (HDM) (Rabie, Saleh & Mansour, 2022), Machine Learning Model (MLM) (Shivahare et al., 2024), and Ensemble Diagnosis Model (EDM) (Shokouhifar et al., 2024). Figures 9–15 illustrate the evaluation of accuracy, error, precision, recall, ROC-AUC, Boxplot, and implementation time. Additionally, Table 3 illustrate the 10-fold cross-validation results for accuracy, precision, and recall. DS outperforms other recent methods.

Figure 9 Accuracy of diagnosis models.

Figure 10 Error of diagnosis models.

Figure 11 Precision of diagnosis models.

Figure 12 Recall of diagnosis models.

Figure 13 ROC-AUC of diagnosis models.

Figure 14 The boxplot of diagnosis models across independent runs.

Figure 15 The implementation time of diagnosis models.

Table 3 The DS’s performance at each fold is based on accuracy, precision, and recall.

Fold#	Accuracy	Precision	Recall	
1	68%	73%	61%	
2	87%	70%	64%	
3	78%	85%	65%	
4	67%	67%	63%	
5	81%	81%	88%	
6	84%	88%	71%	
7	63%	82%	70%	
8	61%	73%	71%	
9	75%	88%	78%	
10	88%	87%	80%	
Average	75.20%	79.40%	71.10%	

Figures 9–15 proved that the proposed DS is better than SNB, HDM, MLM, and EDM. DS introduces the best results at training patients number = 3,444. Figure 9 proved that DS is more accurate than other methods because it provides an accuracy of 89.91%. On the other hand, the error value of DS is 10.09% as shown in Fig. 10. Hence, Fig. 9 showed that the accuracy of DS, SNB, HDM, MLM, and EDM are 89.91%, 65.20%, 70.06%, 77%, and 83.85% respectively. Figure 10 showed that the error of DS, SNB, HDM, MLM, and EDM are 10.09%, 34.80%, 29.94%, 23%, and 16.15%, respectively. Figure 11 illustrated that the precision of DS, SNB, HDM, MLM, and EDM are 73.05%, 55.22%, 65%, 67%, and 72.19%, respectively. The recall of DS, SNB, HDM, MLM, and EDM are 75.14%, 56.08%, 60.98%, 67.29%, and 69.02% respectively as shown in Fig. 12. The ROC-AUC of DS, SNB, HDM, MLM, and EDM are 92.00%, 89.02%, 84.25%, 85.00%, and 87.25%, respectively as presented in Fig. 13. The Boxplot of the diagnosis models after performing 20 independent runs is presented in Fig. 14. According to these results, it is noted that DS represents the best diagnosis method while SNB represents the worst one. Figure 15, the implementation time of DS, SNB, HDM, MLM, and EDM is 8.2, 4.8, 6.89, 5.88, and 6.6 s, respectively. Table 3 demonstrates that the DS can produce the greatest results at each fold as well as average values. Consequently, the DS can obtain the best accuracy, precision, and recall values. Hence, DS can handle the changes in data distribution, class imbalance, or noisy features in real-world medical datasets.

Conclusions and future works

A new DS that includes three main layers called RL, SL, and DL was provided to accurately diagnose cases of various diseases. Outliers were removed by using GA in RL while the best features were selected by using HFSA as a new feature selection method in SL. HFSA consists of two main stages FS and AS. In FS, chi-square, as a filter methodology, is applied in FS to quickly select features while HOA, as a wrapper method, is applied in AS to accurately select features. Finally, the filtered data without outliers or irrelevant features was entered into NB classifier in DL to give accurate diagnoses. Experimental results showed that the proposed HFSA provided the best results according to NB, KNN, and ANN. NB classifier provided more accurate results than KNN and ANN. DS provided the best accuracy, error, precision, recall, and ROC-AUC values which are 89.91%, 10.09%, 73.05%, 75.14%, and 92.00%, respectively. A total of 10-fold cross-validation further validates the DS’s robustness, demonstrating consistent performance across all folds. These results indicate the DS’s ability to handle class imbalance and noisy features, common challenges in real-world medical datasets. While the DS exhibits higher execution time due to the complexity of the preprocessing steps, this is mitigated by the offline nature of these processes, making the system suitable for real-time diagnostic applications. The improved performance of the DS can be attributed to the effectiveness of the HFSA in identifying the most discriminative features, leading to more accurate diagnoses compared to systems employing other feature selection methods.

In the future, building upon the promising results presented in this study, several avenues will be identified. First, the scalability and generalizability of the DS will be rigorously evaluated by testing it on a wider range of datasets. These datasets will vary in size, type (including images, text, and numerical records), and characteristics to ensure the DS can effectively handle diverse data formats and complexities. Second, the RL will be enhanced by integrating a new outlier rejection model that combines novel optimization algorithms with established statistical methods. This aims to further improve the robustness of the system to noisy and inconsistent data. Also, HFSA can be modified by using T2A with GA in a nested loop where T2A is usedto optimize the main parameters of GA to give more accurate results. Finally, and most importantly, the ultimate goal is to translate the theoretical effectiveness of the DS into practical impact. Therefore, future work will focus on implementing the proposed DS within real-world medical and healthcare systems, enabling its deployment as a valuable tool for assisting medical professionals in delivering timely and accurate diagnoses.

Supplemental Information

Supplemental Information 1 Testing dataset.

Supplemental Information 2 Dataset in matlab.

Supplemental Information 3 Total dataset.

Supplemental Information 4 Training dataset.

Supplemental Information 5 Source code.

Additional Information and Declarations

Competing Interests

The authors declare that they have no competing interests.

Author Contributions

Asmaa H. Rabie conceived and designed the experiments, prepared figures and/or tables, authored or reviewed drafts of the article, and approved the final draft.

Mohammed Aldawsari performed the experiments, authored or reviewed drafts of the article, and approved the final draft.

Ahmed I. Saleh performed the experiments, prepared figures and/or tables, authored or reviewed drafts of the article, and approved the final draft.

M. S. Saraya conceived and designed the experiments, performed the computation work, authored or reviewed drafts of the article, and approved the final draft.

Metwally Rashad analyzed the data, authored or reviewed drafts of the article, and approved the final draft.

Data Availability

The following information was supplied regarding data availability:

The datasets and code are available in the Supplemental Files.

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
