# Peer review of "HFSA: hybrid feature selection approach to improve medical diagnostic system"

_PeerJ Computer Science, doi:10.7717/peerj-cs.2764_

## Round 0.1 · original submission · Major Revisions

Dear Authors,

Reviewers have now commented on your article. We do encourage you to address the concerns and criticisms of the reviewers with respect to reporting, experimental design, and validity of the findings and resubmit your article once you have updated it accordingly. Following should also be addressed:

1. A major criticism for the current version of the paper is the lack of justification for the use of the hybrid optimization algorithm combining genetic algorithm and tiki-taka algorithm. Why hybridization is needed and why specifically these two methods are selected for hybridization among many others are not clear.
2. Configuration space of hybrid optimization algorithm should be detailed. It should be more specific and comprehensive. Representation scheme (encoding type) and fitness function with constraint functions should be clearly provided. How constraints (for example: for decision variables) are handled should also be provided. Fitness function should also be provided in detail.
3. Please pay special attention on the usage of abbreviations.
4. Please pay special attention on the usage of referencing style in the text.
5. All of the values for the parameters of all algorithms should be given.
6. Equations should be used with correct equation number. Many of the equations are part of the related sentences. Attention is needed for correct sentence formation.
7. All of the values for the parameters of all algorithms should be given.
8. It is recommended that the paper's experimental results be discussed in greater depth, with additional recommendations and conclusions provided. The conclusion section is lacking in several respects. Firstly, it is essential to describe the academic implications, main findings, shortcomings and directions for future research. Secondly, the conclusion is currently confusing. It is necessary to clarify what will happen next and what we should expect from future papers. To address these issues, the conclusion should be rewritten, taking the following comments into consideration:
- Highlight your analysis and reflect only the important points for the whole paper.
- Mention the benefits
- Mention the implication in the last of this section.
9. Reviewer2 has advised you to provide specific references. You are welcome to add them if you think they are relevant and useful. However, you are under no obligation to include them, and if you do not, it will not affect my decision.

Best wishes,

Reviewer 1 ·

Basic reporting

I have carefully reviewed the revised manuscript, and I would like to provide you with my feedback. The revised manuscript has made some improvements, but there are still several major issues that need to be addressed. Write the comments at the proper place in the manuscript.

1. An exhaustive investigation into the Hybrid Feature Selection Approach (HFSA) is presented in the manuscript. I was wondering if you could explain on the way in which the Fast Stage (FS) and the Accurate Stage (AS) collaborate to improve the predictor selection process.
2. Based on the findings, it can be concluded that HFSA is superior to other methods in terms of accuracy and precision. Regarding the superior performance of HFSA in comparison to FSGA, what specific variables do you believe contribute to this difference?
3. Fitness function is not clear , author should write the fitness function and explore .
4. It is described how long it takes to implement a number of different algorithms. In real applications, how do you think the trade-off between the amount of time it takes to develop something and the accuracy of the algorithm will effect the choice of method?
5. Within the framework of your method, the utilisation of Genetic Algorithm (GA) and Tiki-Taka Algorithm (T2A) is really fascinating. In the context of feature selection, are you able to provide additional insight into the ways in which these computational approaches complement one another?
6. Your approach is intriguing since it makes use of both the Genetic Algorithm (GA) and the Tiki-Taka Algorithm (T2A). In the context of feature selection, are you able to provide additional insight into the ways in which these algorithms augment one another?
7. The paper contains a discussion on the significance of statistical analysis, which can be found by clicking here. In light of the inadequacies that were found in the statistical approaches that were utilised, what steps do you plan to take in order to fix the situation?
8. There is a wide range of precision and recall levels across the various approaches, as demonstrated by the data. What are your thoughts on how these discrepancies should be interpreted in the context of clinical relevance?
9. One of the most important contributions is the Diagnostic System (DS) that has been proposed. In the context of future research in bio-inspired algorithms, what are the potential ramifications that your discoveries could give rise to?
10. The material is well-organised and written in a clear and concise manner. What strategies do you intend to implement in order to ensure that your findings are disseminated to a wider audience operating in the domains of medicine and computation?
11. The research sheds light on the significance of feature selection in terms of enhancing the performance of models. What difficulties did you face when you were going through the process of selecting the features, and how did you manage to overcome them?
12. The research sheds light on the significance of feature selection in terms of enhancing the performance of models. What difficulties did you face when you were going through the process of selecting the features, and how did you manage to overcome them?

Experimental design

Not up to marks, Result should be explore more deeply.

Validity of the findings

The section should be divide clearly:

1. Introduction
List of contribution
2. Related Work
3. Proposed methodology
4. Result and discussion
5. Conclusion
6. References

Current form not acceptable

Additional comments

Explore Proposed methodology ,Result and discussion

Annotated reviews are not available for download in order to protect the identity of reviewers who chose to remain anonymous.
Cite this review as

·

Basic reporting

1. Presentation of the manuscript is excellent
2. The Diagrams presented by the authors are nice per the journal perspective requirement.
3. The literature review is not up to mark.
4. Dataset description is missing in the manuscript even if the authors added it as supplementary data.

Experimental design

1. Why the authors have chosen GA or outlier detection? Why not use another approach? The complexity of GA is high. Address this issue.
2. Why do the authors show interest in the Tiki-Taka Algorithm? Nowhere in the manuscript a detailed description of the importance of this algorithm is mentioned.

Validity of the findings

1. The ROC Curve presentation should be included in the result analysis.
2. Why did the authors use only the NB classifier? Whether they have tried and compared with others. if yes, why not mention it in the manuscript?
3. It is advisable to compare with any two classifiers.
4. Result analysis not done properly. Add the accuracy achieved ( proposed vs existing)
5. Which existing model is compared mentioned clearly?
6. Whether the compared articles use the same dataset? Clarity needed.

Additional comments

The authors are advised to add the details according to review comments.

Reviewer 3 ·

Basic reporting

This paper introduces an advanced Diagnostic System (DS) leveraging Artificial Intelligence (AI) techniques for precise and efficient medical diagnostics. The proposed system incorporates a novel Hybrid Feature Selection Approach (HFSA) that combines filter-based and wrapper-based methodologies to optimize feature selection and enhance the performance of the Naive Bayes (NB) classifier. Through its multi-layered structure—Rejection Layer (RL), Selection Layer (SL), and Diagnostic Layer (DL)—the system demonstrates significant improvements in diagnostic accuracy, precision, and recall. The revision should focus on enhancing the scientific rigor of the methodology, clarifying certain ambiguous details, and strengthening the experimental validation.

a. The novelty and computational feasibility of combining Genetic Algorithm (GA) with the Tiki-Taka Algorithm (T²A) in the Accurate Stage (AS) of HFSA should be elaborated. Include an analysis of why this hybrid optimization is superior to existing methods for feature selection in medical diagnostics.

b. Clarify the choice of chi-square as the filter methodology in the Fast Stage (FS). Provide a justification of its suitability compared to other statistical methods, particularly in handling high-dimensional medical datasets.

c. Expand the explanation of the Rejection Layer (RL) to include details about how GA identifies and removes outliers. Specify whether the definition of outliers is domain-specific or generalized across datasets.

d. Incorporate a thorough evaluation of HFSA against state-of-the-art feature selection approaches. Compare their computational complexity, runtime efficiency, and diagnostic performance to substantiate the claim that HFSA outperforms existing methods.

e. The experimental setup requires more details, including dataset characteristics, parameter configurations, and cross-validation techniques used. This ensures reproducibility and provides transparency for performance metrics such as accuracy, precision, recall, and error rates.

f. The integration of the NB classifier within the Diagnostic Layer (DL) should be critically assessed. Discuss the trade-offs in using NB versus more advanced classifiers like Support Vector Machines (SVM) or ensemble methods in the context of medical diagnostics.

g. Add a detailed discussion of potential limitations of the proposed system. For example, how sensitive is HFSA to changes in data distribution, class imbalance, or noisy features in real-world medical datasets?

h. The paper should include a future research direction section that addresses the scalability of the proposed system, particularly when dealing with rapidly growing datasets or integrating multi-modal medical data (e.g., text, images, and numerical records).

The Literature citation is not adequate, and the related work to machine learning should be discussed:
1. Robust semi-supervised multi-label feature selection based on shared subspace and manifold learning
2. Sparse feature selection using hypergraph Laplacian-based semi-supervised discriminant analysis
3. Nonnegative Matrix Factorization in Dimensionality Reduction: A Survey

Experimental design

not

Validity of the findings

not

Cite this review as

---

## Round 0.2 · accepted · Accept

Dear Authors,

Thank you for clearly addressing the reviewers' comments. Your paper seems sufficiently improved and ready for publication. However, I would be grateful if you could direct your attention to the minor edits proposed by Reviewer 1 prior to proceeding to the production stage.

Best wishes,

Reviewer 1 ·

Basic reporting

In abstract first line, "Thanks to the presence of artificial intelligence methods, the diagnosis of patients can be done quickly and accurately. " This line should be updated.

Experimental design

The author revised the experimental design very nicely.

Validity of the findings

No comment

Additional comments

Author incorporated all the comments very nicely.

Cite this review as

·

Basic reporting

No comments

Experimental design

No comments

Validity of the findings

No comments

Additional comments

No comments

Reviewer 3 ·

Basic reporting

The author has answered satisfactorily the answers of the previous reviewers. The paper is well-written, and the results are sound. The paper deserves to be published.

Experimental design

completed

Validity of the findings

completed

Additional comments

completed

Cite this review as